

# Silencing of the ARK5 gene reverses the drug resistance of multidrug-resistant SGC7901/DDP gastric cancer cells

Hongtao Wan[1,2,*], Xiaowei Liu[1,3,*], Yanglin Chen[1,2], Ren Tang[1,2], Bo Yi[2] and Dan Liu[1]

[1] Jiangxi Provincial Key Laboratory of Basic Pharmacology, Nanchang University School of Pharmaceutical Science, Nanchang, China
[2] Second Abdominal Surgery Department, Jiangxi Province Tumor Hospital, Nanchang, China
[3] Nanchang Joint Programme, Queen Mary University of London, Nanchang, China
[*] These authors contributed equally to this work.

## ABSTRACT

For several years, the multidrug resistance (MDR) of gastric cancer cells has been a thorny issue worldwide regarding the chemotherapy process and needs to be solved. Here, we report that the ARK5 gene could promote the multidrug resistance of gastric cancer cells in vitro and in vivo. In this study, LV-ARK5-RNAi lentivirus was used to transfect the parental cell line SGC7901 and MDR cell line SGC7901/DDP to construct a stable model of ARK5 interference. Subsequently, the cells were treated with four chemotherapeutic drugs, cisplatin (DDP), adriamycin (ADR), 5-fluorouracil (5-FU) and docetaxel (DR) and were subjected to the CCK8, colony formation, adriamycin accumulation and retention, cell apoptosis and other assays. The study found that, in vitro, the expression of ARK5 in MDR gastric cancer cells was significantly higher than that in parental cells. Additionally, when treated with different chemotherapeutic drugs, compared with parental cells, MDR cells also had a higher cell survival rate, higher colony formation number, higher drug pump rate, and lower cell apoptosis rate. Additionally, in xenograft mouse models, MDR cells with high ARK5 expression showed higher resistance to chemotherapeutic drugs than parental cells. Overall, this study revealed that silencing the ARK5 gene can effectively reverse the drug resistance of MDR gastric cancer cells to chemotherapeutic drugs, providing insights into the mechanism of this process related to its inhibition of the active pump-out ability of MDR cells.

## INTRODUCTION

Gastric cancer as a lethal malignancy ranking fifth among cancers in humans, while ranking third in tumor-related mortality worldwide (*Frank-Stromborg, 1989*). Approximately one million people are diagnosed with gastric cancer every year, among whom approximately seven hundred thousand die from the disease (*Karimi et al., 2014*; *Ferlay et al., 2015*). Thus, the current situation of gastric cancer remains severe.

Corresponding authors
Bo Yi, yibo790508@163.com
Dan Liu, liudan1201jx@163.com

In recent years, the preferred treatment for gastric cancer is still chemotherapy and surgery, but neither has effectively improved the survival rate (*Cornejo & Portanova, 2006*; *Kang et al., 2012*) while the five-year survival rate is approximately 20% presently (*Cunningham et al., 2005*; *Isobe et al., 2011*). Additionally, chemotherapy resistance plays a crucial role in the failure of its treatment. Initially, tumors generally respond to chemotherapy, but most of the time they eventually recur (*Linn et al., 1996*) and develop varying levels of cross-resistance to a series of chemotherapeutic drugs after exposure to one drug, and this phenomenon is known as multidrug resistance (MDR) (*Zhang et al., 2011*; *Gottesman, 1993*). Therefore, to improve the sensitivity of tumors to drugs has become a difficult problem in the clinical treatment of gastric cancer while providing an effective entry point for our research.

Studies have demonstrated that tumor cells mainly rely on mechanisms mediated by an efflux pump P-glycoprotein (P-gp) to reduce the accumulation of drugs within cells and alter the subcellular distribution of toxic substances to produce multidrug resistance (*Chabner & Fojo, 1989*; *Biedler & Spengler, 1994*). As a member of the ABC (ATP-binding cassette) family (*Cordon-Cardo et al., 1990*; *Wu & Ambudkar, 2014*; *Xue & Liang, 2012*), the P-gp is reported to be highly expressed in cancer cells (*Goldstein et al., 1989*) and is regarded as the main source of multidrug resistance. However, the latest study found that downregulation of the expression of ABC transporter cannot completely reverse the multidrug resistance of tumor cells, such as gastric cancer cells (*Zhang & Fan, 2010*). Although P-gp is highly expressed in some gastric cancer patients, it was not related to the poor prognosis of patients who were treated with adriamycin and 5-fluorouracil therapy (*Gürel et al., 1999*; *Choi et al., 2002*). Moreover, in a previous study comprising fifty samples of gastric cancer patients, P-gp could only be detected in 10% of them (*Fan et al., 2000*). Thus, some other important pathways exist regarding the occurrence of tumor drug resistance.

Previous studies have indicated that AMP-activated protein kinase (AMPK) is a sensor of metabolism and a regulator of energy homeostasis (*Lu et al., 2019*) and has positive effects on promoting gastric cancer (*Hardie, 2011*). ARK5, a member of the AMPK family, has been suggested to be closely related to cell survival and cell function of tumors (*Chang et al., 2013*). Moreover, the relationships between ARK5 and cancer cells have been verified in many cancer types—for instance, cholangiocarcinoma (*Yu et al., 2017*), hepatoma (*Xu et al., 2016*) and nasopharyngeal carcinoma (*Liu et al., 2018*). The upstream mediator of ARK5 has been identified as AKT (*Suzuki et al., 2004*), LBK1 (*Chang et al., 2013*), CaMKKb (*Hawley et al., 2003*) and TAK1 (*Momcilovic, Hong & Carlson, 2006*). It has been proposed that AKT phosphorylation could stimulate intrusion activity through activating ARK5 to inhibit apoptosis. Meanwhile, ARK5 silencing causes overactivation of the downstream target mTOR, eventually leading to cell death (*Liu et al., 2012*).

According to recent literature, ARK5 was confirmed to be associated with drug resistance in multiple tumors. For instance, the downregulation of ARK5 could significantly increase the chemical sensitivity of lung cancer cells to cisplatin (*Li et al., 2017*). Hepatic cancer cells with high expression of ARK5 showed higher resistance to adriamycin than that of cells with low expression of ARK5 (*Xu et al., 2016*). Similar results have been shown in cases

of cholangiocarcinoma (*Yu et al., 2017*). Thus, we propose that ARK5 plays a key role in the development of drug resistance in tumors. Additionally, ARK5 has been discovered to be closely related to the metastasis and deterioration of gastric cancer cells (*Chen et al., 2017*). However no relevant literature has confirmed the role of ARK5 in the mechanism of multidrug resistance of gastric cancer.

To investigate the molecular basis of the multidrug resistance of gastric cancer, LV-ARK5-RNAi lentivirus was used to transfect the parental cells SGC7901 and MDR cells SGC7901/DDP to construct a stable model of ARK5 interference. Subsequently, the cells were treated with four chemotherapeutic drugs—cisplatin (DDP), adriamycin (ADR), 5-fluorouracil (5-FU) and docetaxel (DR)—and the CCK8, colony formation, adriamycin accumulation and retention, and cell apoptosis assays were performed. We also generated another in vivo xenograft mouse model to explore the relationship between ARK5 and multidrug resistance of gastric cancer cells, and whether interference with the expression of ARK5 protein could effectively reverse the drug resistance of SGC7901/DDP.

## MATERIALS AND METHODS

### Cell culture

The human parental gastric cancer cell line SGC7901 was purchased from the Cell Bank of the Chinese Academy of Sciences (Shanghai, China), and cisplatin-induced multidrug-resistant gastric cancer cell lines SGC7901/DDP were purchased from XiangYa School of Medicine, CSU. The cells were cultured in plates containing the culture medium RPMI-1640 (Solarbio, Beijing, China) with 10% fetal bovine serum (HyClone, America) and double-antibody penicillin-streptomycin. All the cells were grown in a humidified incubator at a constant temperature of 37 degrees with 5% carbon dioxide.

### Transfection

The LV-ARK5-RNAi lentiviruses and negative control (NC) lentiviruses were constructed by Genechem (Shanghai, China). The multidrug-resistant gastric cancer cell line SGC7901/DDP was seeded on 6-well plates and cultured with LV-ARK5-RNAi within the RPMI-1640+HiTransG A transfection system, and then the medium was changed after 12 h followed by incubation for 72 h to achieve optimal interference efficiency for follow-up experiments.

### Western blot analysis

The RIPA lysate (Sigma, America) and PMSF (Sigma, America) were used to lyse and extract proteins at a ratio of 100:1. The concentration of the centrifuged supernatant was measured using a BCA kit (Beyotime, China), and then the protein fluid was mixed with loading buffer (Solarbio, Beijing, China) and denatured at 100 °C for 5 min. The extracted lysates were separated by 10% SDS-PAGE and transferred onto PVDF membrane. Next, the membranes were incubated with primary antibody ($\beta$-actin from Abcam, America; ARK5 form Cell Signaling Technology and Santa Cruz Biotechnology, America) overnight and then were incubated with secondary antibody (anti-rabbit IgG/anti-mouse IgG, ZSGB-Bio, China) for 1.5 h. Finally, protein expression can be observed by chemiluminescence and analyzed using Image lab.
## Cell viability assay

The cell viability was gauged using Cell Counting Kit-8 (TransGen Biotech, Beijing, China). The assays were performed on four different cell groups—parental gastric cancer cells (SGC7901), multidrug-resistant gastric cancer cells (SGC7901/DDP), lentiviruses-transfected multidrug-resistant gastric cancer cells (SGC7901/DDP-shARK5), and negative lentivirus-transfected, multidrug-resistant gastric cancer cells (SGC7901/DDP-NC). All cell groups were treated with cisplatin (DDP), 5-fluorouracil (5-Fu), adriamycin (ADR) and docetaxel (DR) (the drugs were purchased from Macklin, Shanghai, China), and ten groups were set up including the blank group, negative group, and 6 experimental groups with different concentrations of the corresponding drug. After reviewing the literature, we found that the plasma peak concentration (PSC) is a common index in cancer chemotherapy (*Friedman, 1988*; *Burade et al., 2017*; *Dai et al., 2007*), and a previous study (张, *2013*) had successfully applied the plasma peak concentration of four drugs to the cell viability assay of gastric cancer cells. In particular, the four chemotherapeutic drugs were added to the experimental cells of each group at 0.25, 0.5, 1, 2, 4 and 8 times their PSC, and each concentration was repeated in 6 wells. Additionally, the peak serum concentrations of DDP, 5-FU, ADR and DR in the human body are 2.0 $\mu$g/ml, 10 $\mu$g/ml, 0.4 $\mu$g/ml and 0.04 $\mu$g/ml, respectively.

First, cells of different groups were seeded into 96-well-plates and cultured for 24 h to grow adherently. On the second day, the SGC7901/DDP-shARK5 and SGC7901/DDP-NC groups were treated with lentiviruses. Next, four drugs were added to the corresponding experimental group according to different concentration gradients, followed by culture for 48 h (*Zhou et al., 2015*). Thereafter, CCK-8 reagent was added to each well according to the manufacturer's instructions followed by incubation for 2 h away from light. The absorbance was then measured at a wavelength of 450 nm (A450) using a microplate reader (Thermo Scientific, America); the OD measurements were repeated three times and the average was recorded.

## Apoptosis assay

The cells of each experimental group were treated with DDP, 5-FU, ADR and DR according to their corresponding peak serum concentrations and the apoptosis rate was evaluated for each group. First, each group was digested with trypsin (Solarbio, America) without EDTA and then was transferred to a brown EP tube after resuspension and centrifugation twice. After discarding the supernatant, 500 $\mu$l of Annexin binding solution was added to each EP tube to suspend the cells, and 5 $\mu$l of Annexin V-FITC dye (BestBio, Shanghai, China) was added to the binding solution. After mixing, the cells were incubated at 4 °C for 15 min in the dark, and then 10 $\mu$l of PI dye (BestBio, Shanghai, China) was added and incubated for 5 min in the dark. Finally, the apoptotic cells were quantified by flow cytometry (Beckman, America).

## Colony formation assay

The rate of colony formation is the inoculation survival rate of cells and can reflect the proliferation ability of the cell population. First, 800 cells per well were inoculated into

the six-well plate and were cultured at 37 °C for 24 h. Next, four anti-tumor drugs were added to each experimental group according to their peak serum concentrations and then were cultured for 2 to 3 weeks. When a cell clone visible to the naked eye appeared on the plates, the culture was stopped. After washing and drying, the cell was fixed with 4% polymethanol for 15 min. Subsequently, the cells were dyed with 0.1% crystal violet for 20 min, the dye was washed off, and the cells were dried in air. Finally, the colony numbers of more than 50 cells were counted under a microscope, and the colony formation rates were calculated according to the following formula: (Colony formation rate = number of colonies/inoculated cells ×100%).

## In vitro adriamycin accumulation and retention assay

Adriamycin (ADR) is a drug whose fluorescence could reflect the relative content of the drug in cells, indirectly reflecting the ability of the cells to actively pump out chemotherapeutic drugs. First, the cells of each experimental group were seeded into 6-well plates with $5 \times 10^6$ cells per well. Next, according to the literature (张, 2013; Wang, 2004), 5 µg/ml of fluorescent ADR was added to the culture medium of the ADR accumulation group and ADR retention group, while no drug was added to the control group, followed by incubation for 1 h. Specifically, the ADR retention group was washed three times with PBS, followed by incubation with conventional 1640 culture medium for 1 h. Subsequently, the fluorescence of ADR in cells was detected by flow cytometry with an excitation light wavelength of 488 nm and an absorption light wavelength of 575 nm. The pump rate of ADR = (intracellular ADR accumulation - intracellular ADR retention)/(intracellular ADR accumulation) ×100%.

## Xenograft mouse models

Ten 6-week-old female NOD/SCID mice weighing approximately 20 g were purchased from Hunan SJA Laboratory Animal Co., Ltd. and were randomly divided into two groups, five in each group. The mice were kept in SPF animal rooms with five mice in each cage in Nanchang Royo Biotech Co., Ltd. The constant temperature of the environment was 22 °C to 25 °C, and the humidity was approximately 40%. Additionally, the mice were fed with appropriate irradiated feed and sterile water.

5-FU has been widely used as a conventional chemotherapeutic agent for the clinical treatment of gastric cancer (Mahlberg et al., 2017). Therefore, we selected 5-FU, a representative chemotherapeutic drug, to further verify whether the drug resistance of gastric cancer cell lines with high ARK5 expression was higher than that of parental cell lines in vivo. First, during the experiment, SGC7901 cells and SGC7901/DDP cells were digested with trypsin and then were resuspended at $2 \times 10^7$ cells/ml with culture medium. Additionally, the mice were anesthetized with 1% pentobarbital sodium (0.1 ml/20 g) and each was inoculated subcutaneously with $2 \times 10^6$ cells in 100 µl of medium. On the 10th day postinjection, small lumps of the same size could be observed on each mouse, confirming successful inoculation. Thereafter, mice in both the SGC7901 and SGC7901/DDP groups were intraperitoneally injected with anesthetic pentobarbital sodium and 20 mg/kg of 5-FU twice a week for three weeks. Particularly, the weight and tumor size of mice were measured

with a sterile Vernier caliper every three days, and the volumes of tumors were calculated using the following formula: Volume = (length ×widt $h^2$)/2 (*Von Kalle et al., 1986*), where the length and width refer to the maximum and minimum diameters, respectively. On the 31st day postinjection, the mice were weighed for the last time, and then were anesthetized with pentobarbital sodium and sacrificed by breaking the neck. Finally, the tumors were separated from the subcutaneous region of the mice with sterile surgical instruments, and the weight and volume were measured.

All animal procedures were approved by the Laboratory Animal Ethics Committee of Nanchang Royo Biotech Co., Ltd. (Nanchang, China; approval number: RYE2018081801), and all laboratory mice were treated strictly according to the Institutional Animal Care's guidelines in the Guide for the Care and Use of Laboratory Animals published by the US National Institutes of Health (NIH Publication No. 85-23, revised 1996).

## Statistical analysis

The mean ± SEM was used for the data processing of each experimental group. Graphpad Prism 6.0 statistical software was used to analyze the homogeneity test of variance, one-way ANOVA, and least significant difference (LSD). *P* values less than 0.05 were considered to be statistically significant.

## RESULTS

### The ARK5 protein in multidrug-resistant SGC7901/DDP cells is highly expressed.

To investigate the differences in the expression levels of ARK5 protein between parental SGC7901 gastric cancer cells and multidrug-resistant SGC7901/DDP gastric cancer cells, western blot analysis was performed. Compared with the parental cell line SGC7901, the expression level of ARK5 in cisplatin-induced multidrug-resistant cell line SGC7901/DDP was significantly upregulated (Fig. 1).

### Interference efficiency of LV-ARK5-RNAi

After the transfection of multidrug-resistant SGC7901/DDP cells with positive and negative shARK5 lentiviruses, the expression of ARK5 in each group was detected by western blotting. The analysis showed that, compared with multidrug-resistant cells without lentivirus transfection, the expression of ARK5 protein in SGC7901/DDP-shARK5 cells transfected with positive lentiviruses was significantly decreased, while that in SGC7901/DDP-NC cells transfected with negative lentiviruses was unchanged (Fig. 2). This result indicated that this lentivirus transfection system can be used in subsequent experiments.

### Silencing of the ARK5 gene in MDR SGC7901/DDP cells reduces the viability of cells following chemotherapeutic drug treatment

The CCK-8 assay was used to explore the relationship between the ARK5 gene and multidrug-resistant gastric cancer cells. After chemotherapeutic drug treatment, the survival rate of multidrug-resistant SGC7901/DDP cells with high ARK5 expression was significantly higher than that of parental SGC7901 with low ARK5 expression (Fig. 3). However, after the ARK5 gene was silenced by shRNA-ARK5, the survival rate of multidrug-resistant

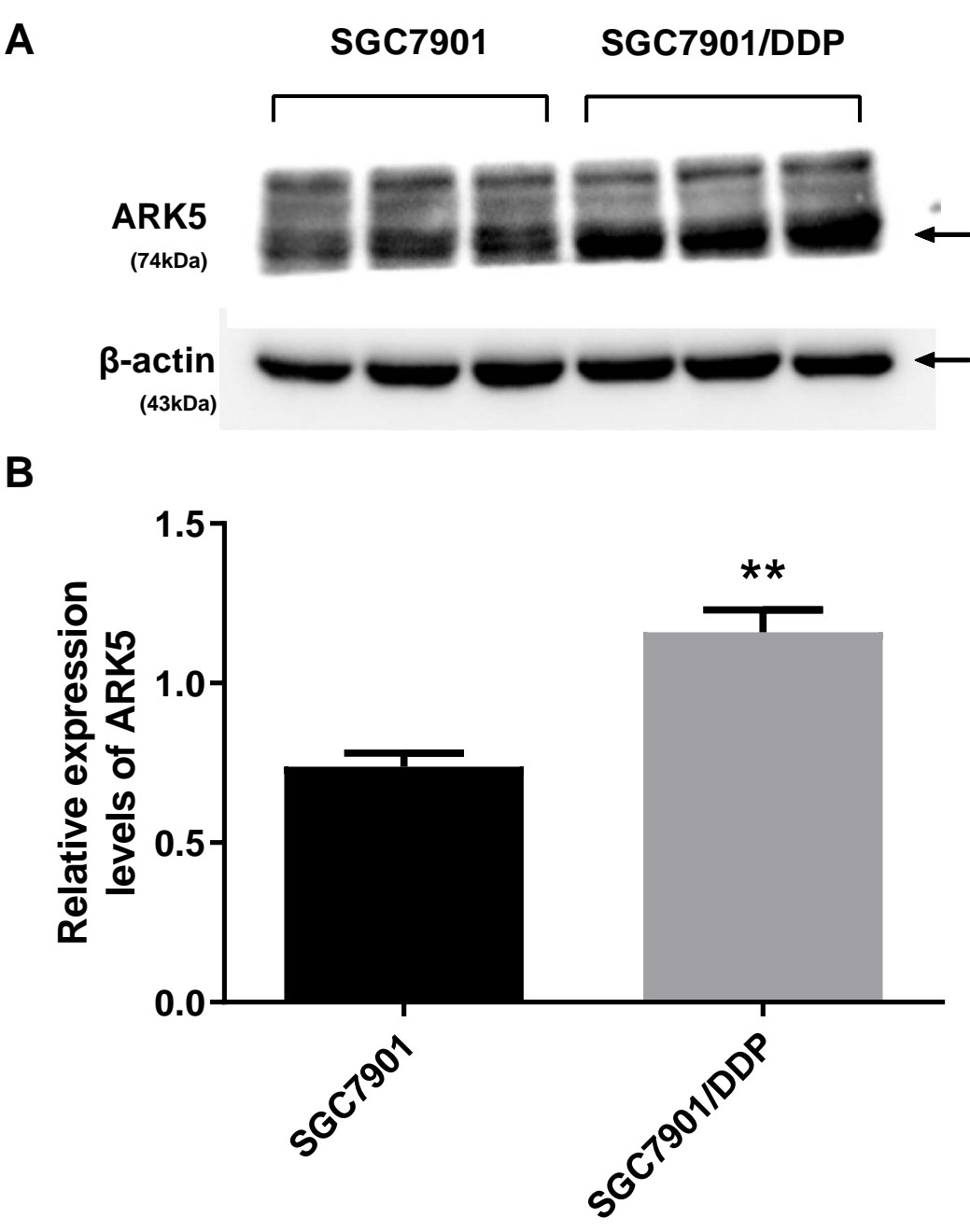

**Figure 1 ARK5 expression levels in parental and multidrug-resistant cell lines.** (A) In this baseline expression level experiment, the protein expression level of ARK5 in SGC7901/DDP was significantly higher than that of SGC7901. (B) The values in a representative blot are shown as the means $\pm$ SEM ($n = 3$; **$P < 0.01$ versus SGC7901).

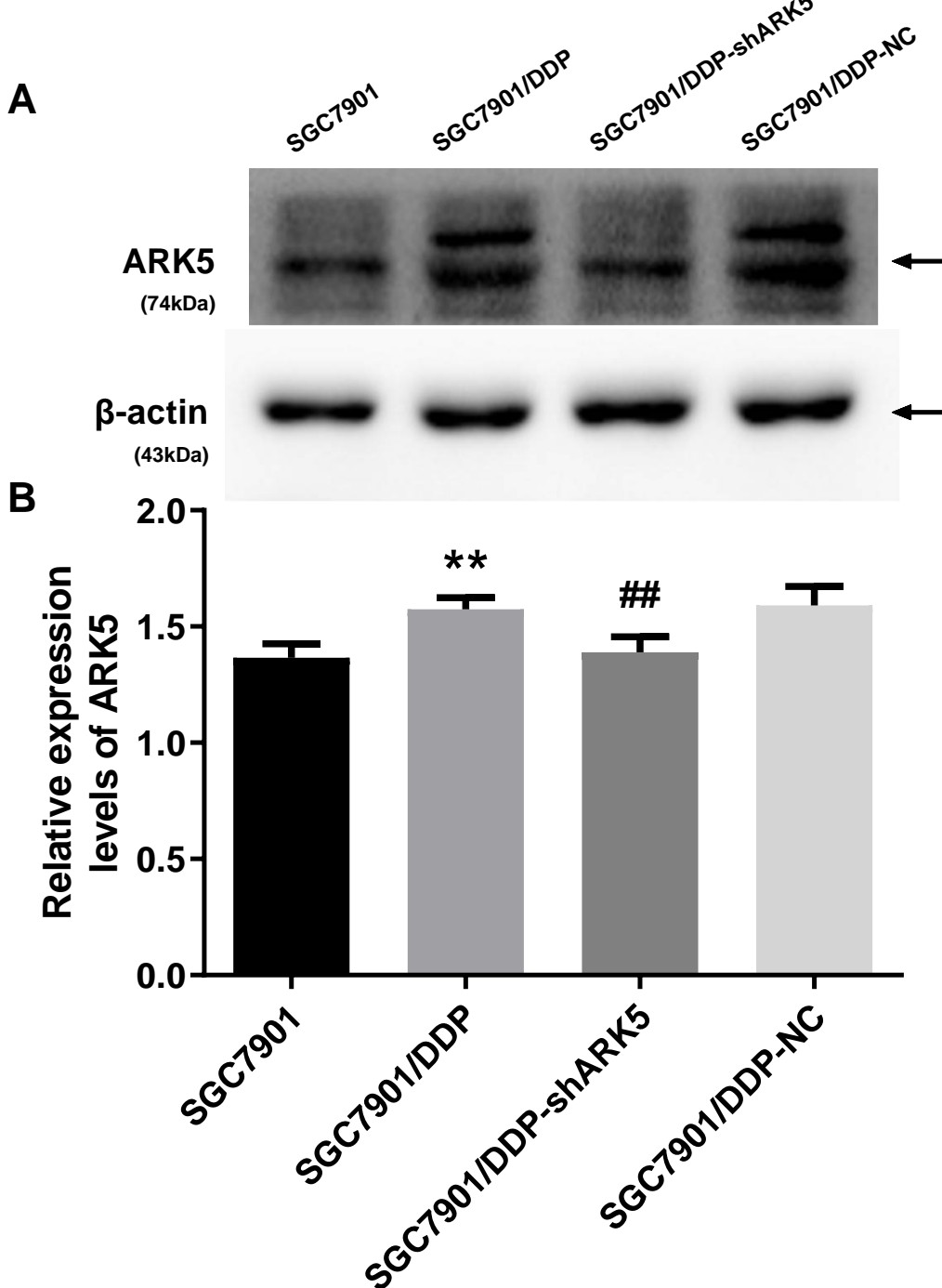

**Figure 2** **Western blot analysis of the interference efficiency of LV-ARK5-RNAi.** (A–B) The differential expression levels of ARK5 in SGC7901, SGC7901/DDP, SGC7901/DDP-shARK5, SGC7901/DDP-NC cells are shown as the means ± SEM ($n = 3$; ##$P < 0.01$ versus SGC7901/DDP; **$P < 0.01$ versus SGC7901). ARK5 expression in SGC7901/DDP-shARK5 was comparable to that of the baseline expression of the parental SGC7901 cell line.

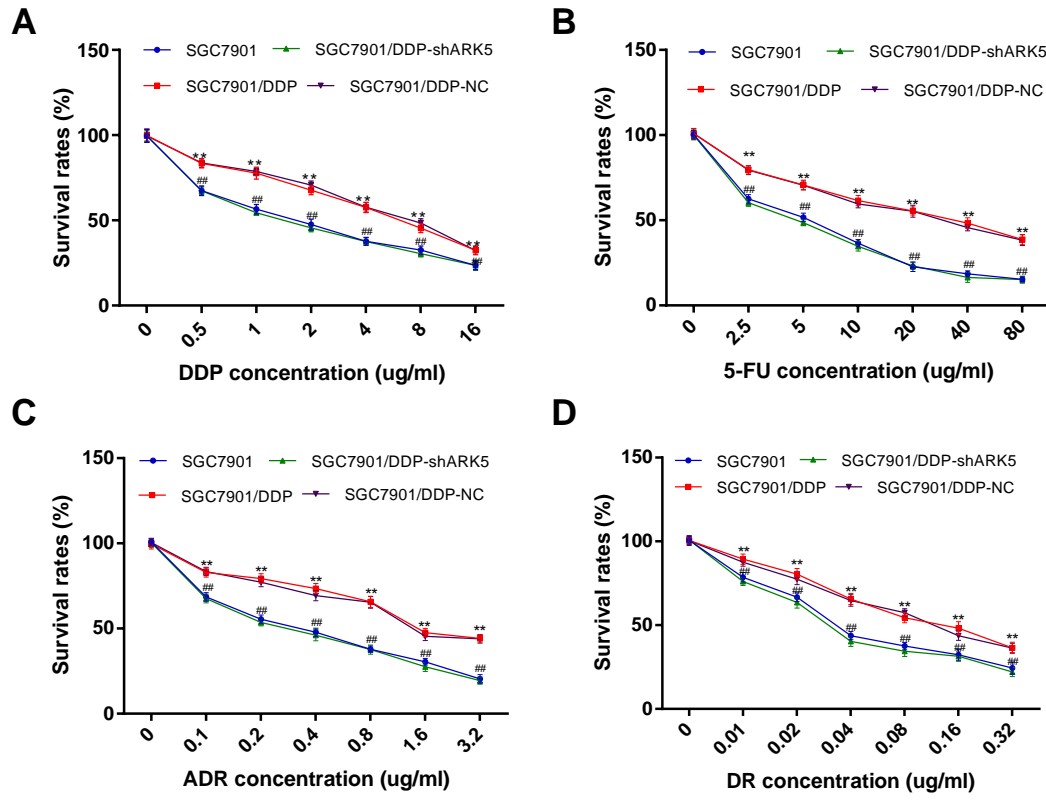

**Figure 3** **Effects of ARK5 gene Silencing on the survival rates in multidrug-resistant gastric cancer cells.** (A–D) After treatment with the four anti-neoplastic drugs (A, DDP; B, 5-Fu; C, ADR; and D, DR), the survival rates of SGC7901, SGC7901/DDP, SGC7901/DDP-shARK5 and SGC7901/DDP-NC cells were measured by the CCK-8 assay. The values are presented as the means ± SEM ($n = 3$; **$P < 0.01$ versus SGC7901; ##$P < 0.01$ versus SGC7901/DDP).

**Table 1** **Drug sensitivity of SGC7901, SGC7901/DDP, SGC7901/DDP-shARK5 and SGC7901/DDPNC cells.** The data are expressed as the means ± SEM; $n = 3$; **$P < 0.01$ versus SGC7901; ##$P < 0.01$ versus SGC7901/DDP.

| Groups | IC$_{50}$ value (µg/ml) | | | |
|---|---|---|---|---|
| | **DDP** | **5Fu** | **ADR** | **DR** |
| SGC7901 | 1.66 ± 0.14 | 5.12 ± 0.38 | 0.38 ± 0.06 | 0.036 ± 0.007 |
| SGC7901/DDP | 6.54 ± 0.36** | 34.28 ± 2.35** | 1.58 ± 0.18** | 0.128 ± 0.016** |
| SGC7901/DDP-shARK5 | 1.42 ± 0.09## | 4.92 ± 0.28## | 0.32 ± 0.09## | 0.031 ± 0.009## |
| SGC7901/DDP-NC | 7.12 ± 0.42 | 32.64 ± 3.42 | 1.42 ± 0.12 | 0.138 ± 0.012 |

cells was significantly decreased compared with that of the normal SGC7901/DDP cells. Additionally, when the transfected lentivirus was negative, no significant change was observed in the survival rate. Meanwhile, the value of IC$_{50}$ (Table 1), which indicates the drug sensitivity of cells, was lower in SGC7901/DDP-shARK5 cells than in normal SGC7901/DDP cells.

### Silencing the ARK5 gene inhibits the proliferation and clonogenic ability of multidrug-resistant SGC7901/DDP cells

After the interference of the ARK5 gene, the number of cell clones in each group was tested by the colony formation assay under different drug treatments. The results showed that (Figs. 4A, 4B), the colony formation rate of multidrug-resistant SGC7901/DDP cells with high ARK5 expression was significantly higher than that of parental SGC7901 cells. After silencing the ARK5 gene, the colony formation rate of SGC7901/DDP-shARK5 cells was significantly decreased.

### Silencing the ARK5 gene in multidrug-resistant GC7901/DDP cells increases chemotherapeutic drug-induced cell apoptosis.

Flow cytometry was used to detect the apoptosis index of cells in each experimental group after chemotherapeutic drug treatment under the condition of ARK5 gene silencing. The results indicated that the apoptosis rate of multidrug-resistant SGC7901/DDP cells with high ARK5 expression was significantly lower than that of parental SGC7901 cells; after silencing the ARK5 gene, the apoptosis index of SGC7901/DDP-shARK5 cells was significantly increased (Figs. 5A, 5B).

### Silencing the ARK5 gene in multidrug-resistant SGC7901/DDP cells decreased the pump rate of Adriamycin in cells

Flow cytometry showed that the adriamycin pump rate of SGC7901/DDP cells with high ARK5 expression was significantly higher than that of SGC7901 cells with low ARK5 expression. However, after silencing the ARK5 gene, the adriamycin pump rate of SGC7901/DDP-shARK5 cells was decreased significantly compared with that of SGC7901/DDP cells (Fig. 6, Table 2).

### The DDP-induced multidrug-resistant cells with high ARK5 expression showed higher resistance to 5-FU than the parental cells in vivo

Based on the results of tumor size from the subcutaneous dissection of mice (Figs. 7A–7X), after multiple administrations of 5-FU, the tumor volume inoculated with multidrug-resistant SGC7901/DDP cells with higher expression levels of ARK5 gene was significantly larger than that of SGC7901 cells (Figs. 5A, 5B). Thus, ARK5 gene expression is proportional to the resistance of gastric cancer cells to 5-FU.

## DISCUSSION

The multidrug resistance of tumors is mainly manifested by the increased resistance and decreased sensitivity of tumor cells and is an indispensable factor influencing the chemotherapy effect in cancer patients. In this study, a series of experiments was conducted to preliminarily explore the relationship between the ARK5 gene and multidrug resistance of gastric cancer cells.

Primarily, western blotting showed that the expression level of ARK5 protein in drug-resistant SGC7901/DDP cells was much higher than that in parental SGC7901 cells, indicating that the drug resistance of cells was associated with ARK5 protein. Next, the

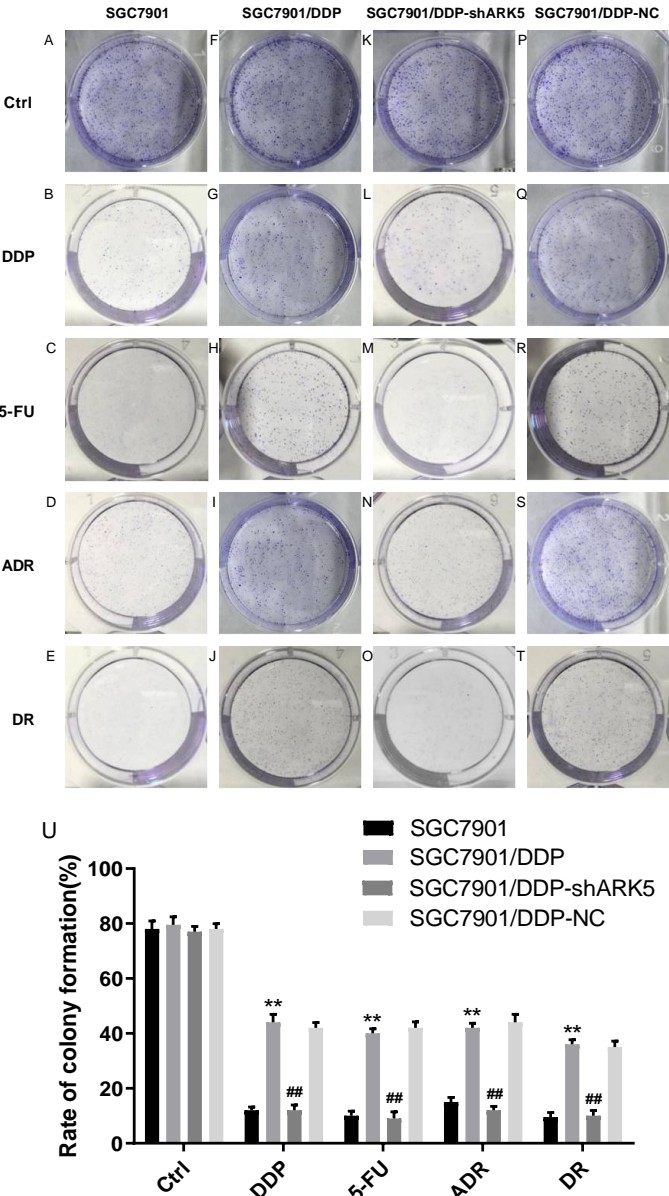

**Figure 4** **Effects of ARK5 gene Silencing on the colony-forming ability in multidrug-resistant gastric cancer cells.** The four anti-tumor drugs DDP, 5-FU, ADR, DR were used to treat each experimental group according to the peak serum concentrations (2.0 µg/ml, 10 µg/ml, 0.4 µg/ml and 0.04 µg/ml, respectively). (A–T) The colony formation assays of SGC7901, SGC7901/DDP, SGC7901/DDP-shARK5 and SGC7901/DDP-NC cells are shown on the plates. (U) The rates of colony formation were measured after treatment with DDP, 5-Fu, ADR and DR. The data are expressed as the means $\pm$ SEM; $n = 3$; **$P < 0.01$ versus SGC7901; ##$P < 0.01$ versus SGC7901/DDP.

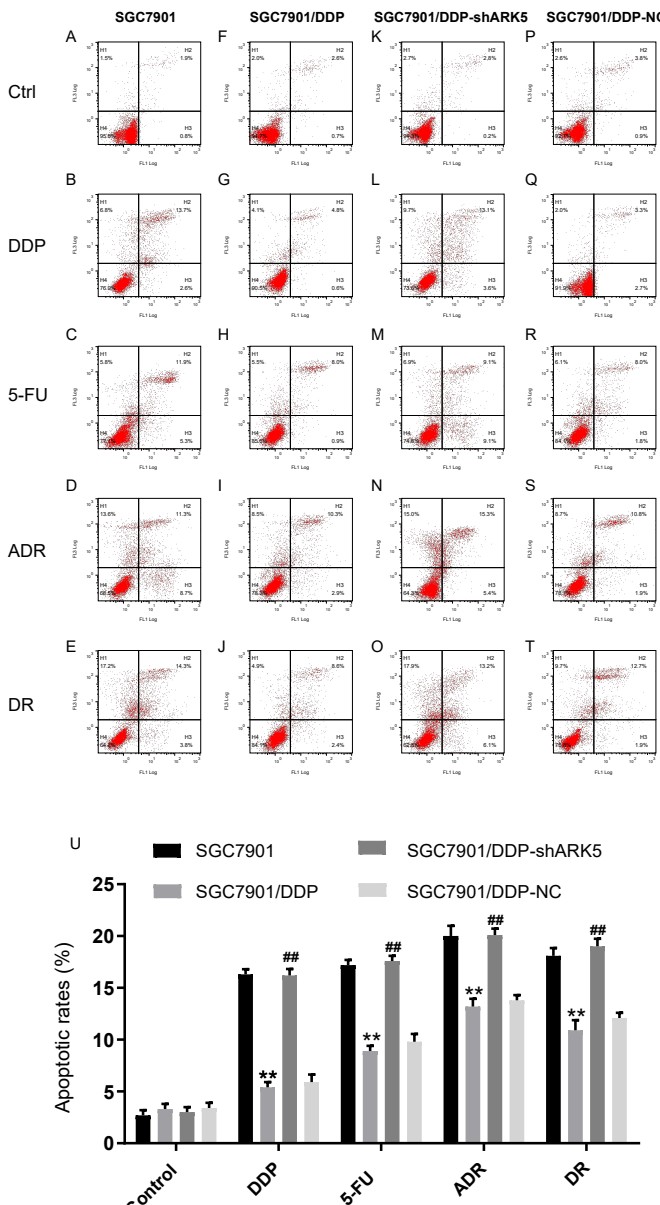

**Figure 5  Effects of ARK5 gene silencing on the apoptosis rate in SGC7901 cells.** The concentrations of the four drugs DDP, 5-FU, ADR and DR used to treat each group were 2.0 μg/ml, 10 μg/ml, 0.4 μg/ml and 0.04 μg/ml respectively. (A–T) The apoptosis rates of SGC7901, SGC7901/DDP, SGC7901/DDP-shARK5 and SGC7901/DDP-NC cells were measured after treatment with DDP, 5-Fu, ADR or DR. The lower left quadrant indicates viable cells, and the other three quadrants represent cells with varying degrees of apoptosis. (U) Bar graph of the apoptosis rate analysis. The data are expressed as the means ± SEM; $n = 3$; **$P < 0.01$ versus SGC7901; ##$P < 0.01$ versus SGC7901/DDP.

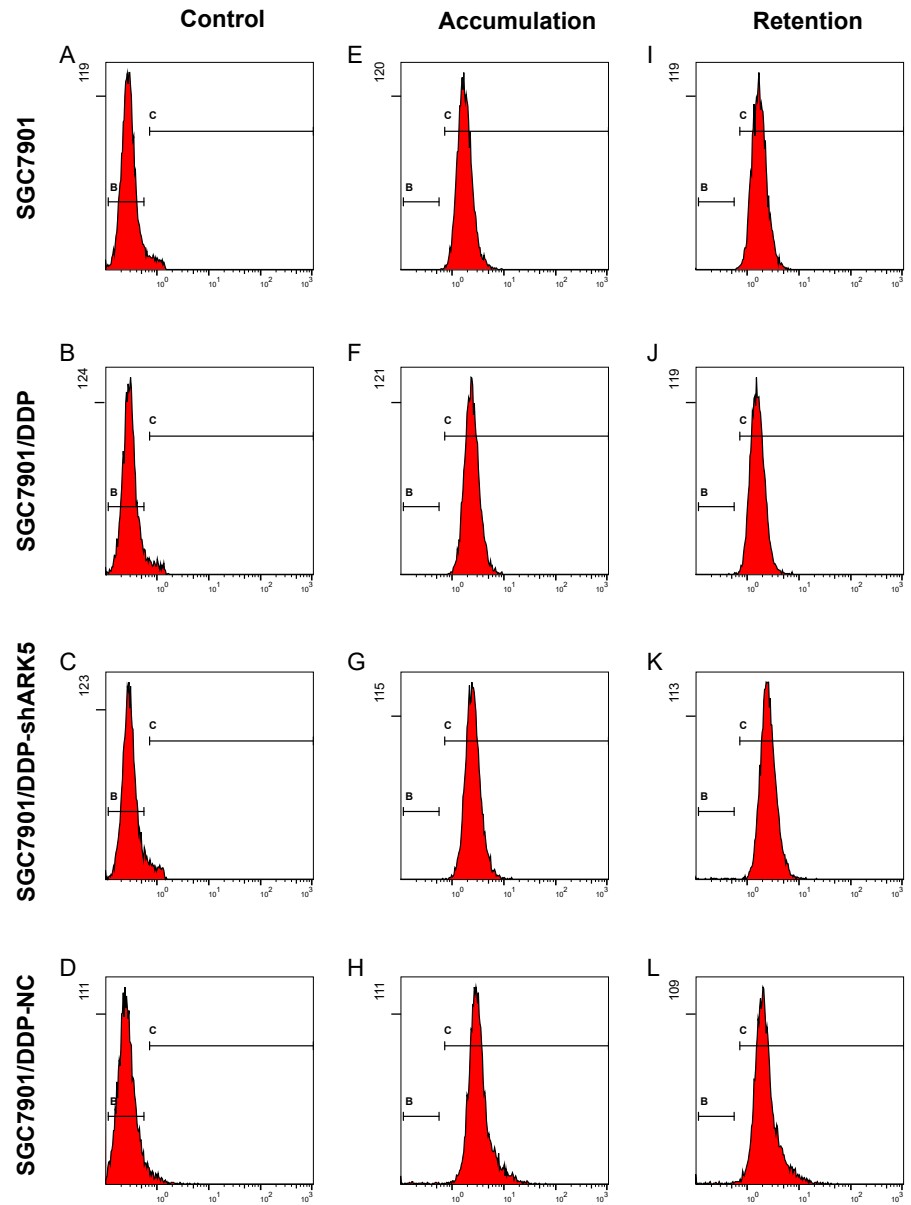

**Figure 6** **Effects of ARK5 gene silencing on the adriamycin pump rate in SGC7901 cells.** (A–L) After cells were treated with 5 μg/ml of adriamycin, the ADR accumulation and retention of SGC7901, SGC7901/DDP, SGC7901/DDP-shARK5 and SGC7901/DDP-NC cells were measured by flow cytometry.

cells in each group were treated with cisplatin (DDP), adriamycin (ADR), 5-fluorouracil (5-FU) and docetaxel (DR). Compared with SGC7901/DDP cells, the low survival rate and low $IC_{50}$ index of SGC7901/DDP-shARK5 cells indicated that interfering with ARK5 gene expression could significantly enhance the sensitivity of cells to anticancer drugs, thus decreasing cell death. Meanwhile, the number of SGC7901/DDP-shARK5 colonies formed was small, indicating that interfering with ARK5 gene expression could significantly reduce the resistance of multidrug-resistant cells to chemotherapy drugs, thus inhibiting the

**Table 2 Accumulation and retention of doxorubicin in gastric cancer cells.** The drug pump rates of SGC7901, SGC7901/DDP, SGC7901/DDP-shARK5 and SGC7901/DDPNC cells were calculated using the following formula: pump rate of ADR = (intracellular ADR accumulation intracellular ADR retention)/(intracellular ADR accumulation) × 100%. The data are expressed as the means ± SEM; $n = 3$; **$P < 0.01$ versus SGC7901; ##$P < 0.01$ versus SGC7901/DDP.

| Groups | Fluorescence Intensity | | Drug pump rate |
|---|---|---|---|
| | Accumulation | Retention | |
| SGC7901 | 1.913 ± 0.086 | 1.864 ± 0.082 | 0.026 ± 0.008 |
| SGC7901/DDP | 2.564 ± 0.184 | 1.713 ± 0.078 | 0.332 ± 0.028** |
| SGC7901/DDP-shARK5 | 2.748 ± 0.092 | 2.701 ± 0.096 | 0.017 ± 0.008## |
| SGC7901/DDP-NC | 3.245 ± 0.204 | 2.264 ± 0.172 | 0.302 ± 0.022 |

proliferation activity and colony-forming efficiency of cells. The above results demonstrated that ARK5 plays an important role in maintaining the resistance of multidrug-resistant gastric cancer cells to chemotherapy drugs and could be effectively overcome by reducing ARK5 expression, thus reversing drug resistance. However, the mechanism remains unclear, and we continued to explore the molecular mechanism of ARK5 in drug-resistant cells.

Increasing the active pump-out ability of anti-tumor drugs and reducing the concentration of the drugs in cells are known to be important ways for cancer cells to develop drug resistance, similar to the ABC transporter family member P-glycoprotein encoded by MDR1 and MDR2 acting as a pump to limit drug accumulation in cells to achieve drug resistance (*Wu & Ambudkar, 2014*; *Xue & Liang, 2012*; *Nieth et al., 2003*; *Abdallah et al., 2016*). In this study, we found that, after silencing the ARK5 gene in drug-resistant cells, the drug pump rate of cells to adriamycin was significantly reduced, indicating that the mechanism of action of ARK5 on multidrug-resistant cells was likely related to its inhibition of the active pump-out ability of chemotherapy drugs. Additionally, an abnormal apoptosis pathway can also induce the multidrug resistance of tumor cells. For example, the resistance of gastric cancer cells to anti-tumor drugs (DDP, 5-FU) was increased significantly after P53 gene mutation (*Menon & Povirk, 2014*; *Matsuhashi et al., 2005*; *Cascinu et al., 1998*). In our apoptosis experiments, we also found that silencing the ARK5 gene can significantly improve the apoptosis rate of drug-resistant cells after treatment with chemotherapy drugs. The above results showed that, at the cellular level, the ARK5 gene silencing effectively overcoming the resistance of drug-resistant cells to chemotherapy drugs is related to the reduction of cell resistance to apoptosis.

Additionally, the results from xenograft mouse models displayed that, after 5-FU treatment, the tumors of multidrug-resistant SGC7901/DDP cells with high ARK5 expression were significantly larger than those of SGC7901, indicating that the higher is the expression of ARK5, the higher is the drug resistance and survival of the tumor.

Notably, this experiment has so far confirmed the basic correlation of ARK5 and drug-resistance of gastric cancer cells, and this correlation is closely related to the active pump-out ability of the cells. Further research into this mechanism may be helpful to expand the clinical solutions to reduce multidrug resistance in gastric cancer cells.

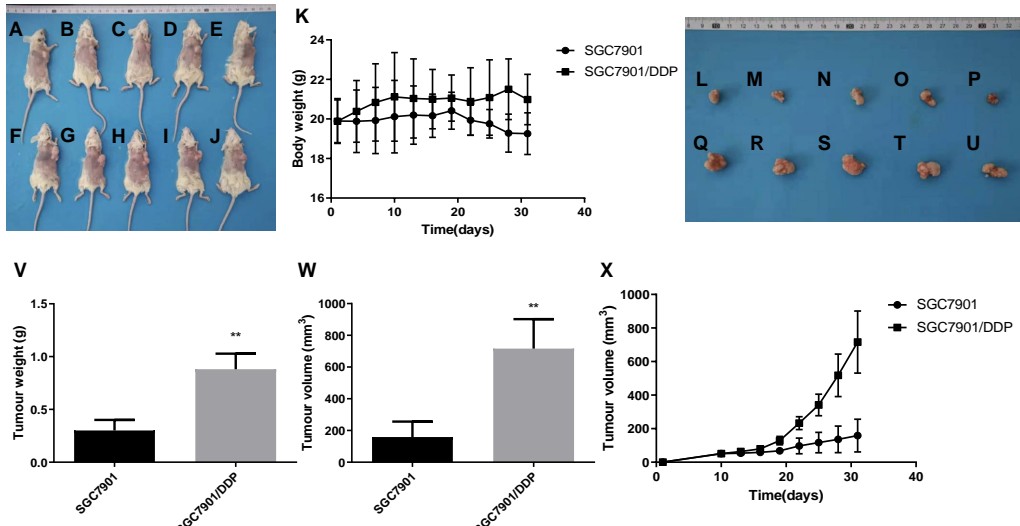

**Figure 7** **Effects of the multidrug resistance of gastric cancer cells on the tumor volume in mice treated with the chemotherapeutic drug 5-FU.** To further confirm that gastric cancer cells with high ARK5 expression have a higher resistance to chemotherapeutic drugs, we used the representative chemotherapeutic drug 5-FU to conduct in vivo experiments in mice and then administered 5-FU at a concentration of 20 mg/kg twice a week. (A-J) Sacrificed and depilated NOD/SCID mice. (K) Body weight variation of mice throughout the 31-day experiment. (L-U) Dissected tumors from mice. (V) Quantification of the tumor weight of mice in the SGC7901 and SGC7901/DDP groups. (W) Quantification of the tumor volume of mice in the SGC7901 and SGC7901/DDP groups. (X) Tumor growth curves of the two groups of mice. The data are expressed as the means $\pm$ SEM, $n = 5$; $**p < 0.01$ versus SGC7901; $^{\#\#}P < 0.01$ versus SGC7901.

In summary, based on the discoveries above, the specific signaling pathways and detailed mechanisms of the drug pumping rate and cell resistance to apoptosis by ARK5 need to be further studied.

## CONCLUSION

Overall, this study verified that the ARK5 gene is closely related to the multidrug resistance of gastric cancer cells in vitro and in vivo, and ARK5 gene silencing could effectively reverse the resistance of multidrug-resistant gastric cancer cells to chemotherapeutic drugs. Moreover, this process is associated with the inhibition of the active pump-out ability of drug-resistant cells and the reduction of cell resistance to apoptosis by silencing the ARK5 gene. Finally, this study provides an experimental basis for the role of ARK5 in the multidrug resistance of gastric cancer cells.

### Funding

This work was supported by the Natural Science Foundation of China (No. 81460551, 81760587, 81460371, 81760731), the Graduate Student Innovation Special Foundation of

Nanchang University (No.cx2016299), and Jiangxi Province Technology Support and Social Development Projects (No. 2010BSA13900). The funders had no role in study design, data collection and analysis, decision to publish, or preparation of the manuscript.

### Grant Disclosures
The following grant information was disclosed by the authors:
Natural Science Foundation of China: 81460551, 81760587, 81460371, 81760731.
Graduate Student Innovation Special Foundation of Nanchang University: cx2016299.
Jiangxi Province Technology Support and Social Development Projects: 2010BSA13900.

### Competing Interests
The authors declare there are no competing interests.

### Author Contributions
- Hongtao Wan and Xiaowei Liu performed the experiments, analyzed the data, prepared figures and/or tables, and approved the final draft.
- Yanglin Chen and Ren Tang analyzed the data, prepared figures and/or tables, and approved the final draft.
- Bo Yi and Dan Liu conceived and designed the experiments, authored or reviewed drafts of the paper, and approved the final draft.

### Animal Ethics
The following information was supplied relating to ethical approvals (i.e., approving body and any reference numbers):

The Laboratory Animal Ethics Committee of Nanchang Royo Biotech Co., Ltd. approved the research (approval number: RYE2018081801).

### Data Availability
The raw measurements are available in the Supplementary Files.

### Supplemental Information
Supplemental information for this article can be found online at http://dx.doi.org/10.7717/peerj.9560#supplemental-information.

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
