# Peer review of "Silencing of the ARK5 gene reverses the drug resistance of multidrug-resistant SGC7901/DDP gastric cancer cells"

_PeerJ, doi:10.7717/peerj.9560_

## Round 0.1 · original submission · Major Revisions

Dear Dr. Liu,

Thank you for your submission to PeerJ. We have conducted peer review of the article titled- "Silencing of ARK5 gene reverses the drug resistance of multidrug-resistant gastric cancer cells SGC7901/DDP by three independent experts in the field. As an academic editor, I am sorry to inform you that the manuscript in its current format does not hold the standards of PeerJ. In accordance with the reviewers' comments, the manuscript in its current form requires Major Revisions. My hope is that you will find the detailed comments of all the reviewers on how to improve the current manuscript useful. We hope to receive your modified manuscript with a response to the reviewers' comments soon. The suggested changes in the manuscript will make it suitable for publication. Below is the summary of the individual comments of each reviewer.

Reviewer 1 ·

Basic reporting

The language used in this article leaves much to be desired. There are several grammatical errors. The language and scientific terminology used are erroneous throughout the manuscript. The authors appear to exaggerate the significance of their findings without adequate scientific data (either from their own study or from available scientific literature) to validate them.

Examples of errors in language/grammar:
Line 26: “ARK5 gene is closely related to the multidrug resistance of gastric cancer cells in vitro and in vivo.” – Language needs to be revised. ARK5 gene could either “promote” or “stimulate” multidrug resistance of gastric cancer cells. The words “closely related” do not have inform the role of this gene in mediating drug resistance.

Line 48 “…satisfactory survival rate…” – what is a satisfactory survival rate? For any disease condition, the obvious satisfactory survival rate is ideally 100%. However, that’s not what we presently observe in cancer therapeutics. Please revise language by citing the current survival rate for gastric cancer.

Line 51: “…exposing to the one drug…” – the correct language here should be “exposing to one drug”.

Experimental design

1. The cell lines used in this study were not authenticated. This substantially brings into question the validity of the findings of this study.

2. Only one set of cell lines (parental and multidrug resistant) were used in this study. The study findings should be confirmed in at least two other cell lines (parental and multidrug resistant) before they can be considered conclusive.

3. In line 172, the authors state, “A total of 6-week-old female NOD/SCID mice…”. The authors need to specify the total number of mice employed in this study.

4. In Figure 2, the western blot image is 1,2, 3 and 4. Please indicate what each of these numbers stand for in the figure legend.

Validity of the findings

The discussion of available scientific literature in this research and therapeutic area is inadequate. Discussion section of this manuscript needs to be revised.

The role of ARK5 in multidrug resistance of gastric cancer cells is not novel. However, the authors claim, "this study not only provides a new experimental basis for the key role of ARK5 in multidrug resistance of gastric cancer cells, but also provides a new target and new exploration direction...". This is not accurate. This study does not provide a "new experimental basis", "a new target", or "new exploration direction" in gastric cancer therapy. The authors repeatedly exaggerate the findings of their study in this manuscript.

Reviewer 2 ·

Basic reporting

The article is clear, written in professional english. The article structure seems to corroborate with the guidelines. Minor mistakes in sentence construction or typos do exist. I have pointed them and provided suggestions. The background seems to be concise and detailed. I have some suggestions which I have mentioned in my report.

Experimental design

Experimental design is sound and well described. The knowledge gap has been addressed and the experimental procedures have been mentioned with sufficient details.

Validity of the findings

Discussion and conclusion are robust, well stated. It is linked to the original research question and supported by the results.

Additional comments

Abstract

Line number 23: “Gastric cancer as…” should be replaced as “ Gastric cancer is…”

Introduction

Line number 43: Please do not start the abstract and the introduction with the same kind of sentence. I would suggest starting the introduction section of the article with a different sentence.

Line number 47-49: Consider breaking up the sentence into two. You can start the second sentence like : “ Besides chemotherapy resistance plays a crucial role in the failure of its treatment.”

Line number 49-52: Consider breaking up the sentence to improve clarity.

Line number 53: Replace “… provides an effective…” with “ ….providing an effective…”

Line number 60: Reference missing.

Line number 60: Consider breaking up the sentence to provide fluidity.

Line number 63: Consider modifying the existing line with: “ Previous studies indicated that AMPK which is a sensor of metabolism and a regulator of energy homeostasis has positive effects on promoting gastric cancer.”

Line number 66-68: Briefly mention what are the upstream and downstream targets of ARK5. Also give brief description of ARK5 pathway.

Line number 72: Add “…cells with high expression of ARK5…”

Line number 73: Replace “ studies” with “ cases”

Line number 74: Consider writing : “We propose that ARK5 may play a….”

Line number 78: Please rewrite this sentence. You can start like “ To investigate the molecular basis ….”

Line number 84: Use a separate sentence. Consider writing “ We also carried out….”

Materials and Methods:

Line 95: Line should be “ …cultured in plates containing the culture…..”.Mention if any antibiotics were used.

Line 162: Mention why this particular concentration of the drug was used for this experiment.

Line 172: “ A total of 10, 6 week old mice…..”. Please mention 10 here.

Line 139: sentence should be “…EP tube after resuspension and centrifugation….”

Results:

Figure 1: Consider writing a Figure caption : “ARK5 expression levels in parental and MDR cell lines”.
Please mention that it is a baseline expression level experiment.
What is the molecular weight of ARK5? Please add a molecular weight marker on the left-hand side and an arrow to denote the protein on the right-side panel of the figure.

Figure 2: Please add a molecular weight marker on the left-hand side and an arrow to denote the protein on the right-side panel of the figure.

Please add that the SGC7901/DDP-shARK5 expression is comparable or almost similar to that of the baseline expression of the parental SGC7901 cell line.

Line 235: Please specify Fig 4 A and B

Figure 4: What was used as a control experiment- DMSO or complete media? Please mention the drug concentrations for each of the drugs that were used in this experiment in the figure legend.

Line number 244: Specify the Fig5 as Fig 5A and 5B.

Figure 5: What was used as a control experiment- DMSO or complete media? Please mention the drug concentrations for each of the drugs that were used in this experiment.

Why the apoptosis rates of SGC7901/DDP in ADR is higher than that of DDP or 5FU drugs? Why more cells are undergoing apoptosis in ADR even though ARK5 is highly expressed? Please clarify this issue.

Figure 6: Mention if DMSO or complete media was used in control experiment in the figure legend?

Figure 7: Mention the name of the drug(s) used in this experiment in the figure legend. Also mention the concentration of the drug(s) in the figure legend.

Figure 7B: The body weight of the SGC7901 parental mice seems to be going down after 20 days. Any reason for that?

Line 258: Please specify the Figure 7 as Figure 7A, 7B …

Line 259: Why only 5-FU was used in this experiment? Why other chemotherapeutic drugs were not tested should be mentioned clearly.

Conclusion:

Line 315: Replace “fin” with “in”

Annotated reviews are not available for download in order to protect the identity of reviewers who chose to remain anonymous.

Reviewer 3 ·

Basic reporting

The article is generally well written, however it needs to be corrections for language and grammar. The references for a study are missing and in addition more literature related to this manuscript needs to be included (as mentioned in comments to author). There are few issues with figure and the required modifications are mentioned in comments to author.

Experimental design

The research question is well defined and relevant to the field. Few experiments needs to be modified for better analysis of the results.
The methods are described fairly, however I have suggested few modifications to improve the manuscript.

Validity of the findings

Although the importance of ARK5 has been reported in lung cancer, but it is a novel study for gastric cancer, which has its own importance.

It seems inclusion of some data related to ABC transporter might shed more light in this study.

Additional comments

Dear Author,

The manuscript entitled “Silencing of ARK5 gene reverses the drug resistance of
multidrug resistant gastric cancer cells SGC7901/DDP” has covered an impact aspect in the field of cancer cell treatment. It has shown the role of ARK5 gene in multidrug resistance (MDR) in gastric cancer and the effect of ARK5 silencing on cancer cell survival.
Overall the manuscript is well written and study well designed, however there are many issues with the scientific rationale and experimental strategies, which need to be considered in order to improve the manuscript.

1. This manuscript has mentioned about ABC (ATP-binding cassette) family, the P-glycoprotein as a factor associated with MDR in various cancers. However, the study focuses on the role of AMP-activated protein kinase (AMPK). Is there any known association between these two pathways and how these might interplay to induce MDR?
2. The authors cited the finding of some latest study as “latest study found that the down-regulation of the expression of ABC transporter could not completely reverse the multidrug resistance of tumor cells”. The reference for this particular study is missing.
3. It should be mentioned in which cancer the ABC transporter was downregulated and insufficient to reverse MDR. The relevance of that particular cancer with gastric cancer must be included, in order to justify the need to study for alternative mechanism in gastric cancer for MDR.
4. The rationale for selecting ARK5 for studying MDR specifically in gastric cancer must be explained. If any specific role of ARK5 in gastric cancer is known that must be included.
5. Possible mechanism of action for ARK5 in regulating MDR in gastric cancer in context of possible downstream molecules might be important for this study.
6. This study predicts the possible role of p53, however the expression levels of p53 have not been estimated.
7. Based on the effect of ARK5 silencing on in vitro adriamycin accumulation and retention assay in treated cells the expression levels of ABC transporter must be analyzed to understand the possible link between these two pathways.
8. The bases for selecting different drug concentration in cell viability assay must be explained.
9. Ideally, the viability assay are done over a period of time like 3-5 days, however it does not seem to have been done likewise. The rationale for choosing a different strategy needs to be included in methods.
10. In methodology section, under apoptosis assay it must be made clear whether or not the cells in supernatant were included.
11. Change the axis title “relative expression level of proteins” to “relative expression level of ARK5” in figure 1 and 2.
12. The western blots for ARK5 in figure 1 and 2 for SGC7901 / DDP and SGC7901 are not consistent, although these are under similar conditions. In addition, there are two bands for ARK5 in figure 2.
13. The manuscript needs to be corrected for English language and grammar.

---

## Round 0.2 · accepted · Accept

Dear Dr. Liu,
Thank you for your resubmission to PeerJ.
I am writing to inform you that your revised manuscript - Silencing of the ARK5 gene reverses the drug resistance of multidrug-resistant SGC7901/DDP gastric cancer cells - has been Accepted for publication.

Reviewer 1 ·

Basic reporting

The authors have revised the manuscript substantially. The revised manuscript is well written.

Experimental design

The authors have revised the Materials and Methods section of the manuscript to address all my comments and concerns.

Validity of the findings

The authors have revised the manuscript to address my previous concerns. No additional comments.

Reviewer 2 ·

Basic reporting

The article is clear, written in professional english. The article structure seems to corroborate with the guidelines. The authors have addressed all the queries I have raised before.

Experimental design

Experimental design is sound and well described. The authors have incorporated all the details required.

Validity of the findings

Validity of the findings have been represented in a clear and concise manner. Based on the reviewers' suggestions they have modified the discussion which has strengthened the impact of this article.

Reviewer 3 ·

Basic reporting

The manuscript has been modified considerably to improve the data interpretation and better undersatnding of the results.

Experimental design

Few experiments those required further clarification and better description of the methods has been achieved.

Validity of the findings

The results and findings in this study have an important scope in this field and seem to be fairly derived.

Additional comments

The manuscript has been improved sufficiently with inclusion of more relevant literature and figures have been also improved.